# Human Cytomegalovirus-Encoded G Protein-Coupled Receptor UL33 Facilitates Virus Dissemination via the Extracellular and Cell-to-Cell Route

**DOI:** 10.3390/v12060594

**Published:** 2020-05-30

**Authors:** Jeffrey R. van Senten, Maarten P. Bebelman, Puck van Gasselt, Nick D. Bergkamp, Jelle van den Bor, Marco Siderius, Martine J. Smit

**Affiliations:** Amsterdam Institute for Molecular and Life Sciences (AIMMS), Division of Medicinal Chemistry, Faculty of Sciences, Vrije Universiteit, De Boelelaan 1108, 1081 HZ Amsterdam, The Netherlands; jrvansenten@hotmail.com (J.R.v.S.); m.p.bebelman@vu.nl (M.P.B.); puckvangasselt@gmail.com (P.v.G.); n.d.bergkamp@vu.nl (N.D.B.); j.vanden.bor@vu.nl (J.v.d.B.); m.siderius@vu.nl (M.S.)

**Keywords:** HCMV, cytomegalovirus, G protein-coupled receptor (GPCR), chemokine receptor, 7TM receptor

## Abstract

Human cytomegalovirus (HCMV) encodes four G protein-coupled receptor (GPCR) homologs. Three of these receptors, UL78, US27 and US28, are known for their roles in HCMV dissemination and latency. Despite importance of its rodent orthologs for viral replication and pathogenesis, such a function is not reported for the HCMV-encoded GPCR UL33. Using the clinical HCMV strain Merlin, we show that UL33 facilitates both cell-associated and cell-free virus transmission. A UL33-deficient virus derivative revealed retarded virus spread, formation of less and smaller plaques, and reduced extracellular progeny during multi-cycle growth analysis in fibroblast cultures compared to parental virus. The growth of UL33-revertant, US28-deficient, and US28-revertant viruses were similar to parental virus under multistep growth conditions. UL33- and US28-deficient Merlin viruses impaired cell-associated virus spread to a similar degree. Thus, the growth defect displayed by the UL33-deficient virus but not the US28-deficient virus reflects UL33’s contribution to extracellular transmission. In conclusion, UL33 facilitates cell-associated and cell-free spread of the clinical HCMV strain Merlin in fibroblast cultures.

## 1. Introduction

Herpesviruses possess genes acquired from hosts via gene capture events which subsequently have been subjected to evolutionary adaptation to support various aspects of the viral life cycle. Some of these viral genes encode for G protein-coupled receptors (GPCRs), the largest class of cell surface receptors in eukaryotes and involved in a wide variety of physiological processes [1]. Viral GPCRs are conserved in all β- and γ-herpesviridae, indicative of their importance in the biology of these viruses [2]. The majority of these viral receptors show homology to cellular chemokine receptors, and several have indeed been demonstrated to bind chemokines and couple to G proteins. Numerous functions have been ascribed to viral GPCRs, including the rewiring of cellular signaling, immune evasion by means of chemokine scavenging and facilitating virus dissemination [2,3].

Human cytomegalovirus (HCMV), the prototypic β-herpesvirus, establishes a latent and usually asymptomatic infection in the majority of the world’s population [4]. In infants with an immature immune system or in immunocompromised individuals, however, this opportunistic virus may cause severe and potentially life-threatening pathological conditions [5]. The HCMV genome encodes four GPCRs, i.e., UL33, UL78, US27, and US28 [6]. Orthologs of UL33 and UL78 are conserved in all β-herpesviruses, whereas US27 and US28 are only encoded by primate CMVs [2]. As all HCMV-encoded GPCR proteins are present on the virion envelope [7,8,9,10], they could influence virion assembly, facilitate virion interaction with target cells, or activate signaling upon deposition on the plasma membrane. Thus far, UL78, US27, and US28 have been reported to contribute to the life cycle of HCMV by facilitating cell entry, transport of virion components to the nucleus, extracellular virion release, cell-associated virus spread, migration of infected cells, and establishment of latent infection [10,11,12,13,14,15,16].

US28 is a constitutively active receptor and its ligand-independent signaling is a prerequisite for establishment of latent HCMV infection [14,15]. US28 furthermore facilitates cell-associated virus spread in epithelial cells, fibroblasts, and vascular smooth muscle cells [11,13]. As a constitutively internalizing receptor able to bind a broad spectrum of chemokines, US28 scavenges chemokines from the microenvironment of infected cells and thereby may contribute to HCMV-mediated immune evasion [17]. Additionally, HCMV-mediated migration of smooth-muscle cells and macrophages depends on chemokine-induced signaling of US28, indicating a role for US28 in HCMV dissemination in vivo [18,19,20]. 

UL33 is an orphan receptor that signals constitutively by coupling to and activating Gα_s_, Gα_i_ and Gα_q_ proteins [21]. Even though UL33’s signaling properties are for the most part similar to US28’s [22], UL33 has not been reported to contribute to the viral life cycle. Several studies demonstrated that HCMV growth in tissue culture is UL33 independent [7,14,21,23,24]. Rodent orthologs of UL33, however, are known virulence factors in their hosts, which argues a similar function for UL33. Although rat CMV (RCMV)-encoded R33 is not essential for virus replication in vitro, R33-null viruses exhibit a replication defect in the salivary gland in rats. Moreover, R33 plays a vital role in RCMV pathogenesis, as disruption of R33 results in reduced mortality of infected rats [25]. Similar observations have been reported for murine CMV (MCMV)-encoded receptor M33. MCMV deficient in M33 displays attenuated growth compared to wild-type virus in primary bone marrow macrophage cultures [26] but replicates like wild-type MCMV in fibroblast cultures [27]. In vivo, M33 facilitates replication of MCMV in spleen, pancreas, and salivary gland [28], which depends on G protein-coupling by M33 in the latter tissue [29]. In addition, establishment of, maintenance of, and/or reactivation from latent MCMV infection in the spleen and lungs depends on G protein-dependent signaling of M33. Recently, constitutive signaling of M33 has been shown to promote migration of infected dendritic cells in lymph nodes, which is required for systemic viral spread [30]. HCMV-encoded UL33 and US28 both complemented the loss of M33. Full complementation is obtained for MCMV replication in spleen and pancreas [26,29], whereas replication in the salivary gland [26,29,31] and reactivation from latency in spleen and lungs [29] are partially rescued by the introduction of UL33 or US28 in the absence of M33. These findings imply conserved functionality and illustrate common functional characteristics of HCMV-encoded GPCRs in vivo.

In this study, we set out to evaluate the role of UL33 and US28 in growth of the clinically relevant HCMV Merlin strain. We report that UL33 is required for efficient viral growth in fibroblasts by contributing to both cell-associated and cell-free transmission. US28 facilitates cell-associated dissemination in fibroblasts, but this receptor is dispensable for efficient multistep viral growth.

## 2. Materials and Methods

### 2.1. Cell Lines and Cell Culture

HFFF TR cells (kindly provided by Richard J. Stanton) were cultured in DMEM (Sigma-Aldrich, St. Louis, MO, USA) supplemented with 10% Gibco FBS (Thermo Fisher Scientific, Waltham, MA, USA), 50 µg/mL penicillin, and 50 µg/mL streptomycin (PPA Laboratories GmbH, Pasching, Austria). Cells were propagated at 37 °C in a humidified atmosphere with 5% CO_2_.

### 2.2. ELISA and Reporter Gene Assays

Expression and signaling of UL33 receptor mutants were assessed in transiently transfected HEK293T cells, as previously described [22]. 

### 2.3. Bacterial Artificial Chromosomes (BAC) Mutagenesis and Recombinant Viruses

HCMV Merlin UL33-HA, ΔUS28, and ΔUL33 bacterial artificial chromosomes (BACs) were described before [22] and derived from pAL1502, kindly provided by Richard J. Stanton. BAC recombineering using *galK* selection [32] was performed to reintroduce UL33-HA or US28 in the ΔUL33 or ΔUS28 BAC, respectively. Adaptations to the protocol, specific for the generation of HCMV Merlin recombinants, are listed. PCR amplicons used to introduce a *galK* expression cassette at the site of UL33 or US28 deletion were constructed using p*galK* (Fredrick National Laboratory for Cancer Research, Frederick, MD, USA) and primers 1–4 (Table 1). To replace *galK* and to reintroduce UL33 or US28, UL33-HA or US28 were amplified from Merlin UL33-HA BAC using primers 5–8. After isolation of BAC DNA, recombinants were checked by means of BamHI or HindIII endonuclease profile analysis and DNA sequencing, as described before [22]. Viruses were produced upon transfection of BAC DNA in HFFF TR cells, as preciously described [22].

### 2.4. Analysis of Virus Growth and Spread

Multi-cycle growth was analyzed upon transfection of 2 × 10^6^ HFFF TR cells with 2 µg BAC DNA. Virus spread was monitored by visual inspection, and micrographs were taken twice a week. Every working day, 3 out of 20 mL supernatant was replaced by fresh culture medium, and three times per week, the 3-mL supernatant samples were cleared from cells and stored at −80 °C for further analysis. HCMV genome numbers were determined as a measure of virus particle numbers released in the supernatant. Following digestion of free genomes using DNAse (M6101, Promega, Madison, WI, USA), virus particles were lysed and genomes were isolated using the QIAamp MinElute Virus Spin Kit (57704, Qiagen, Hilden, Germany), according to the manufacturers protocol. HCMV genomes were quantified by quantitative real-time PCR using primers targeting the glycoprotein B gene; 5′-CTGCGTGATATGAACGTGAAGG-3′ and 5′-ACTGCACGTACGAGCTGTTGG-3′. PCR reactions were performed using SYBR Green Supermix (Bio-Rad, Hercules, CA, USA) with the MyiQ Real-Time PCR detection system (Bio-Rad, Hercules, CA, USA) at 95 °C for 3 min, followed by 40 cycles at 95 °C for 15 s and 60 °C for 1 min.

To monitor cell-associated virus spread, confluent monolayers of HFFF TR cells were infected with 50–100 infectious particles and extracellular spread was prevented through application of a DMEM/1% Avicel RC-591 overlay (FMC corporation, Philadelphia, PA, USA), supplemented with 1 µg/mL doxycycline where indicated. After 14 days of incubation, the overlay was washed away, cells were fixed, and plaques were photographed using a Nikon TE200 microscope (Nikon, Tokyo, Japan), with an Olympus XM10 camera Nikon TE200 microscope (Nikon, Tokyo, Japan). Plaque surface area was determined using Fiji software [33].

### 2.5. Immunofluorescence Microscopy

HCMV-infected HFFF TR cells were fixed 1 and 6 days postinfection. After permeabilization, gB, IE1, pp28, UL33, and US28 proteins were visualized. US28 was stained using rabbit anti-US28 (generated by Covance, Princeton, NJ, USA [34]) and Alexa Fluor^®^ 546-conjugated anti-rabbit antibodies (A11010, Thermo Fisher, Waltham, MA, USA). UL33 was detected using rat anti-HA and Alexa Fluor^®^ 488-linked anti-rat antibodies (A11006, Thermo Fisher Scientific, Waltham, MA, USA). gB, IE1, and pp28 were visualized with respectively mouse anti-gB (ab6499, Abcam, Cambridge, UK), mouse anti-IE1 (MAB810R, Merck Millipore, Billerica, MA, USA), and mouse anti-pp28 (sc-69749, Santa Cruz Biotechnology, Dallas, TX, USA), each in combination with Alexa Fluor^®^ 488-linked anti-mouse secondary antibody (A11001, Thermo Fisher Scientific, Waltham, MA, USA). Cell nuclei were stained using 4′,6-diamidino-2-phenylindole (DAPI) (D9542, Sigma-Aldrich, St. Louis, MO, USA).

Confocal Laser Scanning Microscopy (CLSM) was performed on a Nikon A1R+ microscope (Nikon, Tokyo, Japan) equipped with a 60 × 1.4 oil-immersion objective. Samples were irradiated using the 405, 488, and 561 nm laser lines. The 488 and 561 channels were detected with GaAsP photo-multiplier tubes (PMTs), while the 405 channel (DAPI) was detected with a regular PMT. The samples were scanned with Nikons Galvano scanner (Nikon, Tokyo, Japan) at 2048 × 2048 pixels, corresponding to a pixel size of 104 nm. NIS-Elements AR 4.60.00 software (Nikon, Tokyo, Japan) was used for image acquisition. Fiji was used for image analysis [33].

### 2.6. Multiple Sequence Alignment and Secondary Structure Prediction of UL33

Sequences of the UL33 receptors encoded by different HCMV strains were obtained from AD169, accession number ACL51113.1; FIX, annotated from accession number AC146907; Merlin, accession number ACZ72787.1; Phoebes, annotated from accession number AC146904; TB40/E, accession number ABV71563.1 (adjusted for alternative splicing); Toledo, accession number ASY06300.1; Towne, accession number ACM48023.1; and TR, accession number AGL96633.1. Multiple sequence alignment was performed using Jalview software [35]. The secondary structure of Merlin UL33 was predicted based on multiple sequence alignment of Merlin UL33, AD169 US28 and all human chemokine receptors. 

## 3. Results

### 3.1. Generation of HCMV Merlin Recombinants

Previously, we constructed derivatives of a bacterial artificial chromosome (BAC) clone of the HCMV Merlin clinical isolate [22]. An HA epitope tag was introduced at the C-terminus of UL33 in BAC pAL1502 (Merlin-UL33-HA; Figure 1a) to allow antibody-based detection of UL33. In contrast to N-terminally tagged UL33, C-terminally tagged UL33 is functionally identical to the wild-type receptor, as shown upon ectopic expression of the receptors in HEK293T cells (Appendix A). Next, genes encoding for UL33 or US28 were removed from Merlin-UL33-HA via seamless deletion (Merlin-ΔUL33 and Merlin-ΔUS28, respectively; Figure 1a). While propagating the recombinant viruses derived from these BACs in HFFF TR fibroblasts, repressing the transcription of RL13 and the UL128 locus, a growth defect was observed for the UL33-deficient virus. Based on this observation, we set out to study the role of UL33 and US28 in HCMV Merlin dissemination. To verify that the observed changes in growth characteristics are due to targeted deletion of the genes of interest from the viral genome, UL33-HA or US28 were reintroduced into their respective receptor-deficient mutants. These revertants, named Merlin-UL33rev and Merlin-US28rev, are identical to Merlin-UL33-HA (Figure 1a). Genomic integrity of the Merlin pAL1502 BAC derivatives was validated by qRT-PCR analysis of UL55 (gB), UL33 and US28 genes (Appendix A), endonuclease restriction pattern analysis (Appendix A), and DNA sequencing of the UL33 and US28 gene regions.

### 3.2. UL33 is Important for Virus Spread in Fibroblasts

Multistep growth analysis was performed upon electroporation of the HCMV Merlin-UL33-HA, deletion mutants, or revertants BACs in HFFF TR fibroblasts. Merlin-ΔUL33 exhibited a growth defect compared to Merlin-UL33-HA virus. Virus spread through the monolayer of HFFF TR cells was strongly hampered upon knockout of UL33 as less and smaller plaques were formed in time (Figure 1b). Five days after electroporation, the number of extracellular viral particles in the medium was similar for all recombinant viruses, indicating similar electroporation efficiencies for the BACs (Figure 1c). At later time points, however, the number of Merlin-ΔUL33 virions was lower compared to parental Merlin-UL33-HA (up to 150-fold at day 24). Merlin-UL33rev virus, on the other hand, showed no growth defect, confirming specific recombination of the UL33 gene. Growth properties of HCMV in HFFF TR cells were unaffected by deletion of the US28 gene.

### 3.3. US28 and UL33 Facilitate Cell-to-Cell Virus Spread

US28 facilitates dissemination of the HCMV strain TB40/E via the cell-associated route in fibroblasts [11]. To evaluate the contribution of UL33 and US28 to cell-to-cell transmission of the Merlin strain, confluent HFFF TR cultures were infected at 0.001 infectious virus particles (ivp)/cell and extracellular virus spread was restricted using a semi-solid overlay [36]. Merlin-ΔUL33 and Merlin-ΔUS28 viruses both displayed defects in cell-associated dissemination compared to Merlin-UL33-HA, resulting in 1.6- and 2.2-fold smaller plaques, respectively (Figure 2). Revertant viruses were included as controls and produced plaques of sizes similar to parental Merlin-UL33-HA virus. As a positive control and to place the UL33- and US28-mediated effects in perspective, we used doxycycline to induce the expression of RL13 and the UL128 locus, known to attenuate cell-to-cell spread and multistep virus growth [37] in cultures infected with Merlin-UL33-HA. Under these conditions, plaque sizes were 9.5-fold smaller compared to cultures in which the transcription of the genes was repressed (Figure 2). Together, these results suggest that the loss of UL33’s contribution to cell-associated virus spread is not responsible for the attenuated multistep growth of the Merlin-ΔUL33 virus and argue for involvement of UL33 in extracellular dissemination of HCMV.

### 3.4. Expression of Several Essential HCMV Proteins Is Not Regulated by UL33 

Since UL33 appears to affect cell-free virus dissemination of HCMV, we assessed the impact of UL33 deletion on the expression of several essential HCMV proteins involved in virus production and infection. Expression of envelope protein glycoprotein B (gB or UL55), initiator of viral gene expression immediate early 1 (IE1 or UL123), and tegument protein pp28 (UL99) were not affected in the absence of UL33 in Merlin-ΔUL33-infected cells (Figure 3). Furthermore, the localization of US28, gB, and pp28 to a circular perinuclear compartment suggests that deletion of UL33 does not affect the formation of the Viral Assembly Compartment (VAC).

## 4. Discussion

CMVs can disseminate from infected to uninfected cells via two routes. They can directly spread to neighboring cells with minimal exposure of virions to the extracellular milieu (cell-associated) or via the extracellular release and subsequent cell binding and entry of infectious particles (cell-free). Several viral GPCRs have been implicated to promote dissemination, including three of four GPCRs encoded by HCMV. UL78 and US27 are involved in cell-free transmission, whereas US28 contributes to cell-associated transmission of HCMV [10,11,12,13]. The main finding in this study is that UL33, the fourth HCMV-encoded GPCR, facilitates viral spread via both cell-associated and cell-free transmission. A UL33-deficient derivative of the clinical Merlin strain of HCMV pervaded abnormally slow through a monolayer of HFFF TR fibroblasts, formed less and smaller plaques, and produced less extracellular progeny compared to parental and UL33-revertant virus. For comparison, parental, US28-knock out, and US28-revertant virus grew with similar properties in this multi-cycle growth analysis, which is in line with reports on HCMV AD169, TB40/E, and Towne strains with disrupted or deleted US28 genes [11,23,24,38,39]. Under conditions precluding cell-free transmission, however, cell-associated virus spread was impaired to a similar degree for both viral GPCR-deficient HVMV Merlin virus derivatives. These results demonstrate that UL33 facilitates both cell-associated and cell-free virus spread and that the loss of UL33’s contribution to extracellular transmission causes the growth defect displayed by the UL33-deficient virus.

The HCMV infection cycle is a sophisticated process, involving viral gene transcription, translation of viral proteins, virion assembly, egress of infectious progeny, virus binding and entering a host cell, and translocation of viral components to the nucleus. Which part of the virus replication cycle is impacted by UL33 remains to be elucidated, but we showed that the formation of a VAC and the levels and subcellular localization of IE1, gB, pp28, and US28 are not affected by the loss of UL33. Future efforts should focus on determining whether the release of extracellular progeny or cell entry are impacted by the loss of UL33 as well as whether G protein-mediated signaling is important for UL33’s contribution to virus growth. 

The growth defect exhibited by Merlin-ΔUL33 was clearly apparent by visual inspection of fibroblast cultures during propagation of the virus, whereas previous studies using AD169, TB40/E, and Towne strains reported that UL33 is dispensable for HCMV growth in fibroblasts [7,14,21,23,24]. This dissimilarity of findings might be explained by the setup of the growth assays employed in the different studies. Replication of HCMV Merlin derivatives was evaluated in a multistep growth assay covering a time span of 24 days, whereas several studies assessed viral growth in 5-day single-step assays [7,14,21]. Indeed, the growth defect of Merlin-ΔUL33 was not yet apparent at 5 days pos infection. Similar discrepancies have been reported before; TB40/E deficient of all four GPCRs and US27-deficient AD169 display single-cycle growth properties similar to their wild-type equivalents [14,40], whereas FIX-US27-null virus exhibits a defect in multi-cycle growth [12]. Even though multistep growth assessment is more receptive to reveal growth defects than single-step growth analysis, full-scale profiling of HCMV AD169 and Towne genomes to identify genes relevant for viral replication classified all four GPCR genes as dispensable for multi-cycle viral replication [23,24]. In addition to differences in experimental setup, discrepancies in reported phenotypes might be the result of genetic variation between HCMV strains. As such, HCMV strains differ in viral growth speed, virion infectivity, and preference for cell-free or cell-associated spread [41]. Moreover, compared to low-passage HCMV strains (e.g., TB40/E and Merlin), AD169 and Towne have lost 22 and 19 genes, respectively, due to extensive propagation in fibroblast cultures [42]. Hence, certain HCMV strains might harbor genetic alterations that bypasses their dependency on UL33 for viral growth in fibroblasts. Of note, multiple sequence alignment and secondary structure prediction revealed considerable variation at the N-terminus and extracellular loop 3 between UL33 gene products of different HCMV strains (Figure 4 and Appendix A). These mutations could potentially impact UL33 glycosylation, the constitutive activity of the receptor, as well as the binding of yet unidentified ligands to the receptor. The variability in the UL33 gene further suggests that its functionality might differ from strain to strain. Finally, amino acid differences between UL33 variants should be considered when designing pan-UL33-targeting molecules.

Further research is necessary to explain apparent reported discrepancies regarding the role of UL33 in viral replication of HCMV and potential differences between HCMV strains. Although its single species tropism hampers in vivo characterization of HCMV dissemination, rodent CMVs are well studied in rodent infection models. MCMV- and RCMV-encoded orthologs of UL33, M33, and R33 contribute to viral dissemination and mortality in infected host [25,26,30,31]. Moreover, HCMV UL33 can partially rescue the loss of M33 with respect to in vivo MCMV dissemination [26,29], which argues for a similar role for UL33 in HCMV-infected individuals. Despite current unclarities regarding the role of UL33 in HCMV dissemination, including possible strain-dependent effects, the importance of UL33 for in vitro dissemination, presented here, may advert to exploration of UL33 as drug target for antiviral therapy.

## Figures and Tables

**Figure 1 viruses-12-00594-f001:**
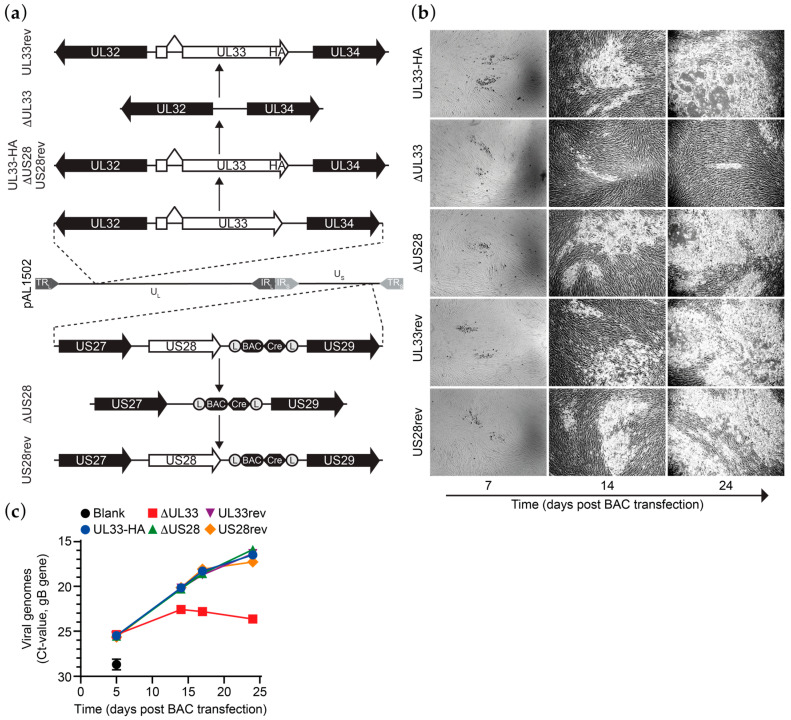
UL33 facilitates Human cytomegalovirus (HCMV) Merlin virus growth in vitro. (**a**) Schematic representation of the recombination strategy to construct HCMV Merlin mutant BACs. Designations of the BAC variations are shown on the left. TR, terminal repeat; IR, internal repeat; U, Unique region; _L_, long; _S_, short; (L), *loxP* sites; BAC, pBeloBAC11; Cre, Cre recombinase; (**b**) micrographs of HCMV virus growth in HFFF TR cells upon reconstitution of the different HCMV Merlin mutants via transfection of the cells with BAC DNA; micrograph size: 2.2 mm × 1.64 mm; (**c**) multistep growth kinetics of Merlin-UL33-HA and mutant viruses, as assessed by the number of secreted virus particles. Results are representative of 3 individual experiments and depicted as mean and SD.

**Figure 2 viruses-12-00594-f002:**
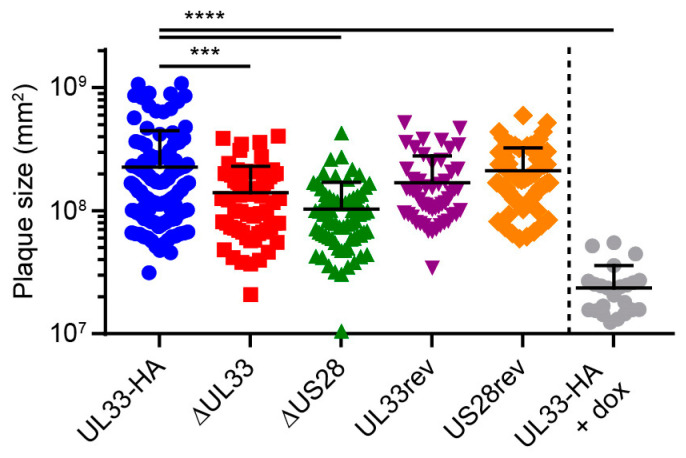
UL33 and US28 contribute to cell-to-cell infection in fibroblasts. Surface area of individual plaques formed by the HCMV Merlin recombinants: HFFF TR cells were infected at 0.001 multiplicity of infection (MOI) and maintained for 14 days under a semi-solid Avicel overlay to prevent cell-free virus spread. Merlin-UL33-HA-infected cells were stimulated with 1 µg/mL doxycycline (dox) to induce expression of RL13 and UL128L. Results from 3 individual experiments were pooled, and error bars show mean and SD. ***, *p* < 0.001; ****, *p* < 0.0001 using Dunnett’s corrected one-way ANOVA.

**Figure 3 viruses-12-00594-f003:**
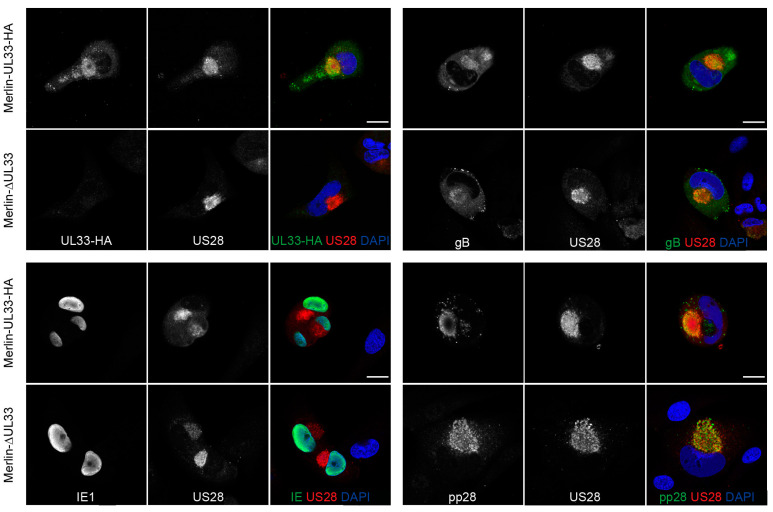
Viral protein expression in HFFF TR cells infected with HCMV-Merlin derivatives: Presence of UL33, US28, IE1, gB, and pp28 at 6 days postinfection of HFFF TR fibroblasts with Merlin-UL33-HA and Merlin-ΔUL33 virus. Scale bar: 20 μm.

**Figure 4 viruses-12-00594-f004:**
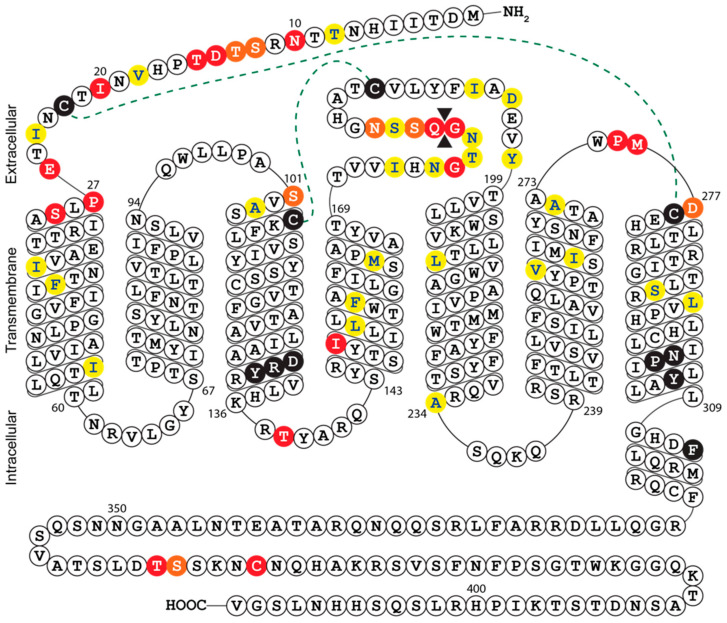
Snake-plot representation of the predicted secondary structure of HCMV Merlin UL33: Conserved residues in chemokine receptors are depicted by black circles and include the DRY motif in transmembrane helix (TM)-III and the NPxxYx_(6)_F motif in TM-VII. Disulfide bridges between cysteine residues in the N-terminus and TM-VII or TM-III and extracellular loop (ECL) II are represented by dotted lines. Amino acid sequence variability of the eight UL33 orthologs is indicated by colored circles; strongly similar properties (e.g., charge, hydrophobicity, and size) are depicted in yellow, weakly similar properties are in orange, and different properties are in red. An amino acid insertion between Gly^179^ and Glu^180^ is depicted by double black triangles. Complete amino acid sequences of the different strains can be found in Appendix A.

**Table 1 viruses-12-00594-t001:** Primers used for bacterial artificial chromosomes (BAC) recombineering and sequencing: Primers 1–4 were used to introduce *galK* (homology arm, sequence recognizing *galK*), and primers 5–8 were used to reintroduce UL33-HA or US28. The oligonucleotides were purchased from Eurofins Genomics, Ebersberg, Germany.

Primer	Primer Description	Primer Sequence (5′-3′)
1	ΔUl33 → *galK* F	CGGAAGCGTCGTCGCCCCGGACTGCGCCCGCGGTCTG CTATTCGTCCACGCCTGTTGACAATTAATCATCGGCA
2	ΔUl33 → *galK* R	GGGAAATGGCGACGGGTTCTGGTGCTTTCTGAATAA AGTAACAGGAAAGCTCAGCACTGTCCTGCTCCTT
3	ΔUS28 → *galK* F	GTGCGTGGACCAGACGGCGTCCATGCACCGAGGGCAG AACTGGTGCTATCCCTGTTGACAATTAATCATCGGCA
4	ΔUS28 → *galK* R	ATCCATAACTTCGTATAATGTATGCTATACGAAGT TATAGCGCTTTTTTATCAGCACTGTCCTGCTCCT
5	*galK* → UL33-HA F	GGAAGCGTCGTCGCCCCGGACTGCG
6	*galK* → UL33-HA R	GGAAATGGCGACGGGTTCTGGTGC
7	*galK* → US28 F	GTGCGTGGACCAGACGGCGTCCATG
8	*galK* → US28 R	CCATAACTTCGTATAATGTATGC

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
