# Peer review of "Human Cytomegalovirus-Encoded G Protein-Coupled Receptor UL33 Facilitates Virus Dissemination via the Extracellular and Cell-to-Cell Route"

_viruses, 2020, doi:10.3390/v12060594_

Round 1

Reviewer 1 Report

This paper by van Senten and collegues investigates the role of the HCMV encoded GPCR UL33 for viral replication. Using recombinant HCMV derived from the Merlin strain they provide evidence that UL33 is important for efficient virus dissemination in cell culture via both the extracellular and cell-to-cell route. These results are interesting and important. However, they are in contrast to previous findings obtained with other HCMV strains. Unfortunately, the authors were not able to explain the reason for these differences. This might be due to differences in the experimental setup which could be investigated by a side-by-side comparison of, for instance, M33 deleted HCMVs derived from Merlin and TB40/E. Indeed, the authors used a quite unusual setup for multistep growth curve analysis: they reconstituted virus by electroporation and immediately monitored virus spread in cell layers. This experimental setting may be affected by differences in electroporation efficacy resulting in different inputs. My recommendation would be to repeat this analysis using virus titrated stocks of the respective viruses.

Reviewer 2 Report

Herpesviral vGPCR proteins play important roles in viral latency, replication and pathogenesis. HCMV encodes four vGPCR proteins, but the precise contributions of these proteins to viral replication remains only partially understood. In this manuscript, van Senten et al., characterize the role of the HCMV UL33 protein in supporting viral replication. This manuscript is clearly written and addresses an important and understudied aspect of herpesvirus biology. The experiments are appropriately controlled and the data is clearly presented. Appropriate statistical analysis is applied throughout.  The methodology is clearly described, including the description of the creation of the mutant and revertant viruses. The authors put their findings in appropriate context of the field.  
